# The Genetic Diversity of Stallions of Different Breeds in Russia

**DOI:** 10.3390/genes14071511

**Published:** 2023-07-24

**Authors:** Natalia Dementieva, Elena Nikitkina, Yuri Shcherbakov, Olga Nikolaeva, Olga Mitrofanova, Anna Ryabova, Mikhail Atroshchenko, Oksana Makhmutova, Alexander Zaitsev

**Affiliations:** 1Russian Research Institute of Farm Animal Genetics and Breeding—Branch of the L.K. Ernst Federal Research Center for Animal Husbandry, 55A, Moskovskoye Sh., Tyarlevo, Pushkin, St. Petersburg 196625, Russia; dementevan@mail.ru (N.D.); yura.10.08.94.94@mail.ru (Y.S.); trantoburito@mail.ru (O.N.); mo1969@mail.ru (O.M.); aniuta.riabova2016@yandex.ru (A.R.); 2All-Russian Research Institute of Horse Breeding (ARRIH), Ryazan Region, Divovo, Rybnovskij District 391105, Russia; atromiks-77@mail.ru (M.A.); 69-mahaon@mail.ru (O.M.); amzaitceff@mail.ru (A.Z.)

**Keywords:** *Equus caballus*, breeds, single-nucleotide polymorphisms (SNPs), genetic diversity, runs of homozygosity

## Abstract

The specifics of breeding and selection significantly affect genetic diversity and variability within a breed. We present the data obtained from the genetic analysis of 21 thoroughbred and warmblood horse breeds. The most detailed information is described from the following breeds: Arabian, Trakehner, French Trotter, Standardbred, and Soviet Heavy Horse. The analysis of 509,617 SNP variants in 87 stallions from 21 populations made it possible to estimate the genetic diversity at the genome-wide level and distinguish the studied horse breeds from each other. In this study, we searched for heterozygous and homozygous ROH regions, evaluated inbreeding using FROH analysis, and generated a population structure using Admixture 1.3 software. Our findings indicate that the Arabian breed is an ancestor of many horse breeds. The study of the full-genome architectonics of breeds is of great practical importance for preserving the genetic characteristics of breeds and managing breeding. Studies were carried out to determine homozygous regions in individual breeds and search for candidate genes in these regions. Fifty-six candidate genes for the influence of selection pressure were identified. Our research reveals genetic diversity consistent with breeding directions and the breeds’ history of origin.

## 1. Introduction

Horse breeding is a branch of animal husbandry in which genotypic variability has a significant range compared with other types of farm animals. Historically, humans have used horses for different purposes, which has led to the genetic diversity of horses. Initially, they served as a source of food, and as they became domesticated, they began to be used as a means of transport [1]. In a number of countries of the modern world, horses are still an important component of food and energy resources.

The variety of modern horse breeds includes breeds that have existed for a long time, as well as relatively young breeds, which were established in the last 100–150 years. At the same time, the intra-breed diversity of horses is not significant. The principles and approaches used in the selection of a particular breed of horse are determined not only by the purposes of breeding but also by geographical and environmental (climatic) conditions. There are different reasons for breeding horses of various uses, both in the past and at present. While economically useful qualities such as productivity and efficiency were of primary significance, in a number of breeds, selection was carried out according to traits and for fixing a certain gait. In many horse breeds, selection has traditionally been carried out according to a set of characteristics. The economic aspects of horse breeding in Russia involve four main areas in which horses are currently being used: working purposes, production, breeding, and sports. In each of these areas, distinct technologies have been developed for maintenance and breeding, as well as the organisation of production; different goals are established; and different strategies are used to increase economic efficiency [2].

The use of molecular genetic technologies has been an important advancement that can increase the efficiency of animal genome evaluation, making it possible to study the genetic architecture of horse breeds. The analysis of microsatellite loci has become widespread in horse breeding, becoming an essential feature of the horse breeding practice for the control of origin and pedigree. It continues to be used to study the genetic diversity and phylogeny [3] of breeds, and new genotyping panels are being developed based on the analysis of repetitive elements in the genome [4] and for the complete sequencing of the horse mitochondrial genome [5]. In 2013, the Equine Genetic Diversity Consortium (EGDC) published the results of Illumina 50 K SNP Beadchip genotyping of 814 samples from 36 breeds [5]. This previous report considers the associations between breeds, which largely reflect the geographical origin and the known history of the breed. Recent studies extract data on SNP genotypes from the 700 thousand Affymetrix Axiom™ Equine Array, which has made it possible to search for genomic associations and study the structure of populations with high resolution [6,7,8,9,10,11]. In many investigations, researchers use genome scanning for overlapping homozygosity (ROH) sequences and for searching these regions for candidate genes using gene ontology [12,13,14].

The aim of this work is to reveal the features of the whole-genome architecture obtained using data from more than 500,000 SNPs in horses of various origins and areas of use.

## 2. Materials and Methods

### 2.1. Ethics Statement

The principles of laboratory animal care were followed, and all procedures were conducted according to the ethical guidelines of the L.K. Ernst Federal Science Center for Animal Husbandry. The protocol was approved by the Commission on the Ethics of Animal Experiments of the L.K. Ernst Federal Science Center for Animal Husbandry (Protocol Number: 2020/2) and the Law of the Russian Federation on Veterinary Medicine No. 4979-1 (14 May 1993).

### 2.2. DNA Extraction

DNA was isolated from the sperm of stallions stored in the Cryobank of Genetic Resources (FGBNU All-Russian Research Institute of Horse Breeding, Ryazan Region, which is part of the Network Bioresource Collection of Farm Animals, Birds, Fish, and Insects). DNA was collected from 87 stallions of 21 breeds. For this study, samples were obtained from the following breeds: Arabian, Akhal-Teke, Terek, Hanoverian, Trakehner, Holstein, Don, Budyonnovskaya, Thoroughbred, Rhine, Oryol, French Trotter, American Standardbred Trotter, Vladimir Draft, Russian Draft, Soviet Draft, Welsh Pony, French Sel, Draft Crossbreed, and Half-breed. Figure 1 shows examples of the individual stallions used in the study.

DNA was isolated from stallion sperm according to the standard method using mercaptoethanol to destroy the membranes of spermatozoa, proteinase K to remove proteins, and phenol extraction. The quality of the obtained DNA samples was determined in several stages: (1) by measuring the concentration of double-stranded DNA on a Qubit™ (1.0) fluorimeter (Invitrogen, Life Technologies, Waltham, MA, USA); (2) by determining the ratio of the degree of absorption of DNA preparations at 260 and 280 nm using a NanoDrop 8000 device (Thermo Scientific, Waltham, MA, USA); and (3) by determining the degree of DNA degradation via electrophoresis in agarose gel with a concentration of 1%.

### 2.3. Genotyping DNA Samples

DNA genotyping was performed on Axiom™ Equine Genotyping Array chips (Thermo Fisher Scientific, USA). DNA samples with a genotyping quality of more than 98% for SNP loci were selected. SNP selection was performed using the PLINK 1.9 program [15]. After quality control, 509,617 SNP variants were used for further analysis.

### 2.4. Statistical Analysis and Visualisation

The genomic architecture of horse breeds was calculated using the program Admixture 1.3 [16] and graphically visualised in the R package pophelper of the R Studio 1.3.1093 software [17]. The most probable number of ancestral clusters (K) was determined by calculating cross-validation (CV) errors for K values from 2 to 6.

Genetic diversity was assessed based on the expected (He) and observed (Ho) heterozygosity values calculated using the PLINK 1.9 program [18].

The calculation of inbreeding based on FROH was carried out using the R package detectRUNS. The minimum ROH length size was 500 kb to minimise the number of false-positive results [19].

## 3. Results

The analysis of over 50,000 SNP variants revealed the genomic architecture of 21 horse breeds. The cross-validation error is shown in Figure 2. The smallest error value of 0.49676 was determined at K = 3. The studied groups of horses were analysed considering the number of ancestral clusters equal to 3. In this case, clear differentiation was observed for the group of American Standardbred (AMST) horses, which formed a distinct cluster without traces of other breeds (Figure 3). With this value of K, it was also possible to distinguish separate groups of horses of the Arabian (AR) and Soviet Heavy Draft (SH) breeds. 

At K = 4, the Trakehner breed was identified as a cluster with similarity to the Thoroughbred breed. This commonality persists even as the number of probable ancestors increases. The French Trotter (FR), with a significant number of ancestral populations, stood out from the rest, demonstrating commonality with the American Trotters (AMST) and the Thoroughbred (ENG). 

A highly informative picture of the horse population structure is displayed at K = 5 (cross-validation error 0.50464).

Figure 4 shows that most French Trotters are characterised by a low average individual level of inbreeding, but there are individuals that can be characterised as outliers. A high level of inbreeding is observed in Standardbred horses.

Since some horse breeds were represented by a low number of individuals in the total sample, five populations were selected for further calculations, from which at least six samples were obtained for analysis. Heterozygosity values for the selected horse breeds are presented in Table 1. The highest expected heterozygosity value was noted for the American Standardbred breed (0.346), and the lowest for the Arabian breed (0.290).

For each of the studied breeds, the maximum number of homozygosity runs was observed on different chromosomes (Figure 5). Thus, for the Arabian breed, a slightly higher ROH was observed on chromosomes 12, 15, and 27. The graph clearly shows that the population of this breed has a great number of outliers. In representatives of the French Trotter breed, the majority of ROH was found on chromosome 7, whereas representatives of the Standardbred breed did not contain homozygosity runs on this chromosome. On chromosome 22, most of the ROH islands were noted in representatives of trotting horse breeds and the Soviet Heavy Draft. 

Further annotation of genes was carried out in homozygous regions. Table 2 shows the homozygosity hotspots associated with horse breed characteristics. Notably, 14 homozygous regions were found, occurring in all the studied representatives of four of the five horse breeds. The largest number of homozygous loci (seven) was found in the American Standardbred on chromosomes 4, 7, 8, and 18. Three homozygous regions were found in Trakehner horses on chromosomes 3, 5, and 17. Three homozygous regions were found in Soviet Heavy Draft horses on chromosomes 3 and 17. One homozygous region was found in the French Trotter on chromosome 22

## 4. Discussion

In horse breeding, the use of information provided in studbooks to describe the origin of animals is widespread [20]. Unfortunately, it is practically impossible to avoid errors in pedigrees, especially when working with old data. This can lead to incorrect estimates of inbreeding [21], which increases the possibility of producing undesirable offspring when selection is based on studbook only.

The development of genetic technologies, including genome-wide screening using SNP data, has provided a good alternative to methods based on pedigree analysis. Using custom-designed chips, inbreeding levels can be compared both between breeds [6] and within individual groups [22]. Analysis chip data also make it possible to search for traces of selection in the genomes of various horse populations.

In our study, it was possible to analyse representatives of different breeds. These are breeds for which using producers of other breeds is not allowed, the so-called purebred and half-breeds. The analysed populations also differed in the type of occupancy.

The population structure and genetic homogeneity of the studied sample of horses were calculated using the Admixture 1.3 program. (Figure 3). The graph clearly demonstrates that the group of Standardbred individuals (n = 6) forms a distinct cluster already at K = 3. In the French Trotter group, at K = 5, there are traces of the influence of the Standardbred and Thoroughbred, which corresponds to the history of the origin of this group of horses. At K = 6, traces of the genome of the Orlov Trotter breed appear, which can also be explained by the origin of the breed. However, the very nature of these common genome fragments is not a marker for the Oryol Trotter horses. Other researchers have investigated the genetic characteristics and population structure of the Asil, Caspian, Dareshuri, Kurdish, and Turkmen horses, which are local Iranian horse breeds [23]. 

The Arabian breed was characterised by low heterozygosity compared with other horse groups considered in this study, as evidenced by the values calculated for all SNPs (E (hom) = 0.290, O (hom) = 0.293). This is slightly lower than the values reported by Cosgrove et al., (2020) [24], who analysed 378 Arabian horses from 12 countries, and in their study, heterozygosity values ranged from 0.30 to 0.33. This is probably due to the peculiarities of breeding Arabian horses with a limited number of animals, combined with the breeders’ desire to carry out heterogeneous selection. The population of Arabian horses considered in this study formed a separate group (Figure 3), but the individuals within it were highly heterogeneous. This is evidenced by the FROH values (Figure 4 and Figure 5), along with the large number of animals included in the analysis that can be described as “outliers”. 

FROH revealed a significant range of values for the group of Arabian horses (Figure 4 and Figure 5). Since the extended homozygous segments develop in descendants from parents with identical haplotypes, the length of these segments will indicate the likelihood that inbreeding occurred relatively recently [22,23,24,25,26,27,28]. Arabian horses have mostly short runs of homozygosity, which may indicate the presence of common ancestors in the distant ranks of the pedigree. The analysis of the distribution of homozygous regions across chromosomes did not reveal the sites that could be assigned to the regions susceptible to selection. The genome of the Arabian horses we studied has unique features characteristic of the breed, and at the same time, no extended ROH was observed. This pattern of ROH distribution probably reflects the breeding characteristics of Arabian horses. The genetic diversity of Arabian horses in Russia is based on the use of different breed types in breeding (Koheilan, Koheilan-Siglawi, Siglawi, and Hadban).

Based on the results of ROH analysis, the inbreeding of the Standardbred breed had high values compared with other breeds, despite their unrelated origin. The slightly scattered values and consolidation of FROH on chromosomes 11, 14, 21, 24, 28, and 31 indicate a non-random distribution of the islands of homozygosity and the effect of selection pressure on the breed (Figure 4 and Figure 5). Following the results of Esdaile (2022) [27], genomic inbreeding based on the proportion of the genome covered with ROH (FROH) ranged from 12% to 27% (average F ROH = 21%), which is less than the result of our study. This is due to not including short ROHs ranging from 500 to 1000 Mb in the assessment, which account for most of the homozygosity runs that determine “ancient” inbreeding. FROH islet analysis of chromosomes revealed differences between trotting, riding, and heavy draft breeds. The high occurrence of FROH in trotting and draft horses may be related to the narrow selection direction, as opposed to saddle breeds. 

In our study, DNA samples of Arabian horses from different regions were analysed. All animals were categorised into the same cluster, which indicates the same trend of breeding practices as those in different countries.

Figure 3 shows the influence of the Arabian breed on the Oryol Trotter, as well as on half-breeds, originally bred in the European region (Hanoverian, Holstein, and Rhine). All these groups had traces of Arabian horse genomes, which indicates that the Arabian breed has long been used for improving local breeds in Europe as early as the late 19th and early 20th centuries [28]. 

In the course of analysing the structure of horse populations, it was found that, already at K = 3, horses of the Trakehner breed have many genomic features in common with Thoroughbred riding horses. Historically, the Thoroughbred has been a major influence in Trakehner breeding for 250 years. In the 1940s, the Trakehner breed underwent a significant reduction in livestock, and the current animals of this breed are descendants of a limited number of producers. In a survey conducted on four German warmblood breeds [29], ROH searches led to the identification of 149 runs of homozygosity in Trakehner horses, whereas only 5 and 39 ROHs were found in Holstein and Hanoverian breeds, respectively.

Hanoverian and Holstein horses were bred for use in agriculture and later for military purposes. Therefore, despite a certain influence of the Thoroughbred horse breed, they retained traces of the genomes of the Arabian breed and trotting breeds to a greater extent than the Trakehner breed (Figure 3). 

Along with the Arabian, Thoroughbred, and Standardbred breeds, heavy horses also play a significant role. Cluster analysis showed a high differentiation of horses of the Soviet Heavy Draft breed from other groups of horses. This is well explained by the origin of the breed, which was officially registered in 1952. When creating the breed, mainly local Russian draft-type horses of various origins, bred via absorptive crossbreeding with Brabansons, were used as a maternal basis. Representatives of other draft breeds in our study were used as individual specimens but have traces of common origin with the Soviet Heavy Draft.

For the study of breeds, it is important to identify specific areas of homozygosity associated with selection.

Two homozygous regions were found on chromosome 3, the occurrence of which in the Soviet Heavy Draft population was 100%. The loci included the genes presented in Table 2.

The PIEZO1 gene, which has been studied in humans and mice, contributes to iron metabolism [30]. Other studies have found that the expression of this gene affects resistance to the loss of bone mass and bone resorption [31]. The high growth rate and selection of massive heavy breeds likely resulted in the accumulation of homozygosity in the 106.22–107.84 region on chromosome 3. Other authors have described the association of genes from this region with high daily weight gain in steers [32].

In our study, the ROH islet, including the QDPR and FAM184B genes, was found in the Soviet Heavy Draft horse breed, while in the work of Szmatoła et al., it was observed only in draft horses [33]. A homozygous region comprising the FGF5 gene, which are mutations that are associated with coat length in donkeys, has been found in American Standardbred horses [34] and is associated with the disease degenerative suspensory ligament desmitis [35].

Another gene, namely CFAP299 (from the same region), has also been associated with the coat length of white yaks [36]. More research is needed in this region regarding American Standardbred horses to establish the causes and consequences of the accumulation of homozygous variants.

Genomic regions were found to be accumulated homozygous regions on chromosome 4 in all studied American Standardbred horses (n = 8). Part of the genes annotated at these sites perform tasks related to general cellular function. Thus, the NUP42 gene is responsible for the synthesis of the protein nucleoporin 42, which promotes the export of protein molecules from the nucleus [37]. The STK3 gene produces the serine–threonine protein kinase 3 and provides the ability of cells to resist unfavourable environmental conditions [38]. The most interesting gene associated with chromosome 4 is the IL-6 gene. Interleukin 6 stimulates energy mobilisation in muscle and adipose tissue and is thought to be a myokine, a cytokine produced by muscle, whose levels increase in response to muscle contraction. It is assumed to act similarly to hormones during exercises by mobilising extracellular substrates and/or increasing their supply.

Interleukins act as intracellular communication signals in haematopoiesis, stress, inflammation, immunity, and tissue repair [39]. Studies investigating the regulation of IL-6R expression in horses state that the system controlling the inflammatory response in horses is better adapted to physical activity than in humans [40]. American Standardbred horses are bred and used for racing in hippodromes. It can be hypothesised that the accumulation of ROH in a region involving a pleiotropic cytokine, which is produced in response to exercise or tissue damage, is not random and is associated with the adaptation of horses to significant muscular exertion. A homozygous region was found on chromosome 5 in all the studied animals of the Trakenen breed, presumably associated with mental ability in horses. The *AMIGO1* gene has been previously linked to the genetic basis of cognitive function in humans [41].

Chromosome 7 exhibited a large extended homozygous region in American Standardbred horses (Table 2). It includes many genes, some of which deserve particular attention. BARX2 is an important regulator of muscle growth and repair that functions by controlling satellite cell proliferation and differentiation [42]. The TMEM45B gene produces the body’s hypersensitivity to mechanical pain caused by inflammation and tissue damage [43]. Other authors have found candidate genes in thoroughbred horses that have been implicated in muscle development, among them ADAMTS15 [44], which we also found in a homozygous region in all American Standardbred horses. NTM and OPCML genes are regarded as significant genes in the genomic selection of thoroughbred horses for predicting racing participation [45], and they are also used in the selection of other horse breeds [46]. IgSF9b regulates anxious behaviour through effects on centromedial inhibitory synapses of the amygdala [47]. Studies in thoroughbred horses have identified a candidate gene JAM3, involved in spermatogenesis in horses and genes CALR and DNMT1 mediating fertility and pregnancy in mares [48,49]. The dysfunction of Condensin II protein in humans affects mitochondrial respiration and mitochondrial oxidative stress response [50]. Other researchers have found that the PRDX2 gene plays a functional role in oxidative stress in equine species; JUNB, PIN1, and KANK2 genes are involved in repair processes in ligament and tendon injuries; and LDLR regulates lipid metabolism, which affects the risks of degenerative suspensory ligament desmitis disease [51]. The ACAD8 gene in horses is assumed to influence the muscle tissue structure, as SNPs of the ACAD8 gene in cattle are used as an indicator to improve the growth rate and tenderness in beef [52]. The GIPC1 gene is suspected to affect muscle motor function, which has an association with movement disorders in humans when CGG repeats are expanded [53].

IL-27 has an important metabolic function in humans and is a promising target for immunotherapy against obesity [54]. In American Standardbred horses, it may be associated with a musculature trait. The RLN3 gene is thought to be involved in eating disorders in humans and mice [55].

The CACNA1A gene is involved in muscle contraction in many horse breeds [32]. In our study, we found a homozygous locus in this region in both American Standardbreds and French Trotter horses.

The ACP5 gene protein participates in osteogenesis and is associated with selection for bone strength in trotting horse breeds [56]. The ECSIT gene is associated with heart muscle development, and a low level of human ECSIT protein leads to mitochondrial dysfunction and cardiac pathophysiology [57]. The TMEM241 gene was found on chromosome ECA8 in AMST in the homozygous region, with a frequency of occurrence of 100%. This gene is associated with bone degeneration and osteoporosis in humans [58]. The presence of a homozygous state of the annotated gene may be attributed to directional selection pressure to improve endurance and athletic performance.

The RIOK3 gene located at ECA8 in AMST is responsible for the innate immune response in humans [59]. The presence of this gene in the homozygous state is probably explained by the history of the American Standardbred breed and its selection for resistance to infectious diseases.

One homozygous region was found on chromosome 11 in the Soviet Heavy Draft, which includes a group of HOXB family genes (HOXB1, HOXB2, HOXB3, HOXB4, HOXB5, HOXB8, HOXB9, and HOXB13). They play a fundamental role in animal morphological diversity and the control of axial morphology along the anteroposterior axis of the body [60].

The SNX11 gene affects body thermoregulation in mice and humans [61]. The presence of these genes in the homozygous region may be caused by selection for endurance among heavy breeders.

The NFE2L1 gene on chromosome ECA11 is associated with lipid metabolism and obesity in humans and mice [62]. A large amount of brown fat is a typical feature of heavy-duty horse breeds.

The CBX1 and ABI3 genes in humans are related to neurocognitive development [63], and the TAC4 gene relates to movement coordination in humans and mice [64].

The CALCOCO2 and NXPH3 genes, which are involved in degenerative joint disease in humans, have been discovered in a Soviet heavyweight at ECA11 [65]. The presence of a homozygous state of this gene may be due to selection for endurance in heavy horses.

A gene belonging to the ZNF gene family, which plays a key role in the regulation of muscle growth and differentiation, was also found in ECA11. The discovered gene may be associated with selection pressure for strength, endurance, and a high percentage of muscle tissue, especially in draft breeds [33].

A homozygous region of length 21,063,873–22,280,900, including the genes associated with muscle, ligament, and tendon development (TRIM13, SPRYD7, and SETDB2), was found at ECA17 in horses of the Trakenen breed [66], which influences the persistence of inflammatory joint disease along an ontology structure similar to humans (EBPL) [67], the presence of cognitive abilities (KPNA3 and RCBTB1) [68,69] and immune response (PHF11 and CYSLTR2) [70,71].

The presence of annotated genes in the homozygous state in the Trakenen breed is probably related to selection for strong and athletic qualities in this horse breed. In French trotting horses, ROH was found in ECA22 with a frequency of occurrence of 100%. The genes included in this homozygous region are mainly associated with the development of different muscle groups (GPCPD1), different metabolic processes in the body (FERMT1, CRLS1, and CHGB) [72] and behavioural traits (PROKR2) [73,74].

The accumulation of homozygous variants of these genes is probably related to the history of the origin of the French trotting breed (purebred breeding) and selection for endurance and persistence.

## 5. Conclusions

The analysis of 509,617 SNP variants in 87 stallions from 21 populations allowed us to assess the genetic diversity at the genome-wide level and differentiate the studied horse breeds from each other. Thus, the whole-genome architecture of breeds is of great practical importance for preserving the genetic characteristics of breeds and managing selection. ROH islet analysis has proven to be a valuable approach to identifying differences between highly specialised breeds (such as trotting and draft breeds) and riding breeds. Our studies revealed genetic diversity corresponding to the direction of breeding and the history of the origin of the breeds. The heterogeneity of the occurrence of homozygous regions, and the low overall heterozygosity of horses of the Arabian breed with a slight level of inbreeding, were noted. Horses of the Trakehner breed showed genetic similarity to Thoroughbreds. Representatives of the Soviet Heavy Draft breed formed a separate cluster while showing their influence on other heavy draft breeds included in the analysis.

Studies were carried out to identify hotspots of homozygosity associated with the breed characteristics of horses. Notably, 14 homozygous regions were found, occurring in all the studied representatives of four of the five breeds of horses. The American Standardbred had a high level of inbreeding and a significant occurrence of regions of homozygosity on chromosomes 4, 7, 8, and 18. “Traces of selection” were identified, and 56 candidate genes significant for selection were annotated. In horse breeding, it is important to consider the accumulation of homozygosity to reduce the effects of inbreeding.

## Figures and Tables

**Figure 1 genes-14-01511-f001:**
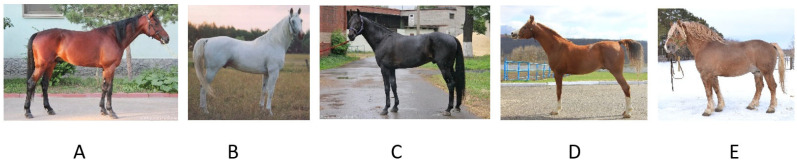
Examples of individual stallions used in the present study: (**A**) Kvantum, of the American Standardbred Trotter; (**B**) Volf, of the Trakehner breed; (**C**) Forvard Loc, of the French Trotter; (**D**) Ecspirien, of the Arabian breed and (**E**) Rasstrel, of the Soviet Draft.

**Figure 2 genes-14-01511-f002:**
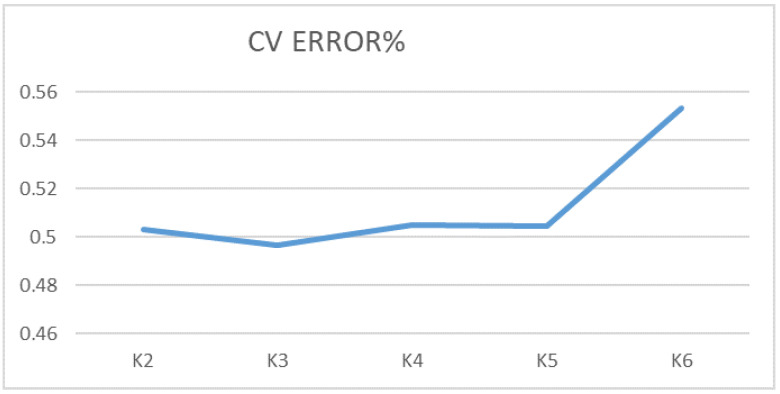
Graph obtained when calculating the number of ancestral clusters (K) from 2 to 6 based on cross-validation errors (CV% error).

**Figure 3 genes-14-01511-f003:**
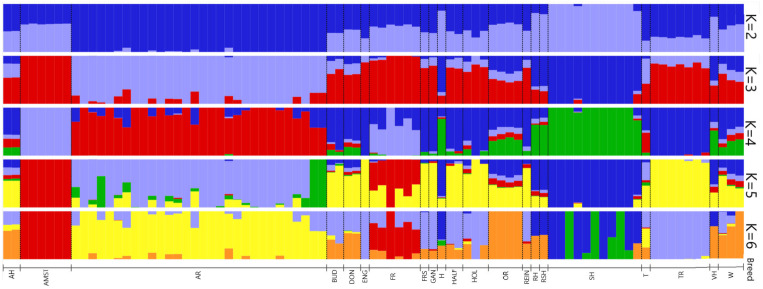
Comparative structure of horse populations calculated using the program Admixture 1.3 with the number of ancestral clusters K from 2 to 6. Each bar represents an individual animal for each terminal breed and the six colors represent each K population cluster.

**Figure 4 genes-14-01511-f004:**
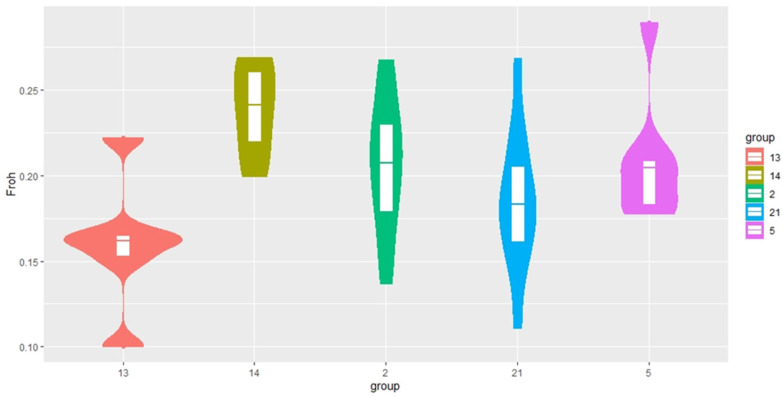
Plot of mean, quartiles, and frequency (plot width) based on the ROH of the coefficient of inbreeding (FROH) for each breed group. Breed designations: 2—Arabian (AR); 5—Trakehner (TR); 13—French Trotter (FR); 14—Standardbred (AMST); 21—Soviet Heavy Draft (SH).

**Figure 5 genes-14-01511-f005:**
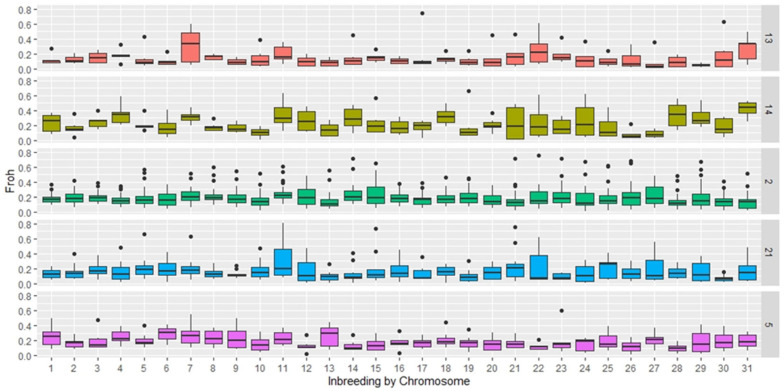
Inbreeding calculated from FROH for five horse populations by chromosome. Breed designations: 2—Arabian (AR); 5—Trakehner (TR); 13—French Trotter (FR); 14—Standardbred (AMST); 21—Soviet Heavy Draft (SH).

**Table 1 genes-14-01511-t001:** Values (M ± SE) of the observed heterozygosity O (hom) and expected heterozygosity E (hom) calculated from all SNPs for the five horse populations.

Breed	Number of Samples	Observed Heterozygosity O (hom)	Expected Heterozygosity E (hom)
Arabian (AR)	30	0.293 ± 0.003	0.290 ± 0.000
Trakehner (TR)	6	0.351 ± 0.007	0.336 ± 0.000
French Trotter (FR)	7	0.375 ± 0.007	0.344 ± 0.000
American Standardbred (AMST)	6	0.375 ± 0.006	0.346 ± 0.000
Soviet Heavy Draft (SH)	11	0.324 ± 0.006	0.313 ± 0.000

**Table 2 genes-14-01511-t002:** Annotated candidate genes on ROH hotspots in Trakenen, French Trotter, Standardbred, and Soviet Heavy Horse. Breed designations: 5—Trakehner (TR); 13—French Trotter (FR); 14—Standardbred (AMST); 21—Soviet Heavy Draft (SH). ECA—*Equus caballus* chromosome number.

ECA	Region (Mb)	Breed	Genes
3	36.02 … 36.03	SH	*PIEZO1, CTU2, CDT1, APRT, GALNS*
	57.96 … 57.97	AMST	*FGF5, PRDM8, CFAP299, ANTXR2, GK2*
	106.22 … 107.84	SH	*NCSPG, DCAF16, FAM184B, LAP3, QDPR, CLRN2*
4	20.22 … 20.23	AMST	*IKZF1, FIGNL1, DDC*
	54.16 … 55.71	AMST	*IL6, TOMM7, FAM126A, KLHL7, NUP42, GPNMB, MALSU1, IGF2BP3, TRA2A, CCDC126, FAM221A, STK31*
	71.23 … 71.73	AMST	*PPP1R3A, FOXP2*
	93.10 … 93.92	AMST	*HIPK2, TBXAS1, PARP12, KDM7A, SLC37A3, RAB19, MKRN1, DENND2A, ADCK2, BRAF*
5	55.16 … 56.17	TR	*GNAI3, GPR61, AMIGO1, ATXN7L2, SYPL2*
7	39.15 … 51.80	AMST, FR	*BARX2, TMEM45B, NFRKB, PRDM10, APLP2, ST14, ZBTB44, ADAMTS8, ADAMTS15, SNX19, NTM, OPCML, SPATA19, IGSF9B, JAM3, NCAPD3, ACAD8, THYN1, GLB1L2, B3GAT1, ADGRE3, CLEC17A, NDUFB17, TECR, DNAJB1, GIPC1, PTGER1, PKN1, DDX39A, ADGRE5, ASF1B, PRKACA, C7H19orf67, PALM3, IL27RA, RLN3, DCAF15, RFX1, PODNL1, CC2D1A, BRME1, NANOS3, ZSWIM4, C7H19orf53, MRI1, YJU2B, CACNA1A, IER2,STX10, NACC1, TRMIT1, LYL1, NFIX, DAND5, GADD45GIP1, RAD23A, CALR, FARSA, GCDH, KLF1, DNASE2, MAST1, RTBDN, RNASEH2A, PRDX2, THSD8, JUNB, HOOK2, BEST2, GET3, TRIR, TNPO2, FBXW9, GNG14, DHPS, WDR83, WDR83OS, MAN2B1, ZNF791, ACP5, ELOF1, CNN1, ECSIT, ZNF653, ELAVL3, PRKCSH, RGL3, EPOR, SWSAP1, PLPPR2, TMEM205, CCDC159, RAB3D, TSPAN16, ANGPTL8, DOCK6, KANK2, SPC24, LDLR, SMARCA4, TIMM29, YIPF2, CARM1, C7H19orf38, TMED1, DNM2, QTRT1, ILF3, SLC44A2, AP1M2, CDKN2D, KRI1, ATG4D, S1PR5, KEAP1, PDE4A, CDC37, TYK2, ICAM3, RAVER1, ICAM5, ICAM4, ICAM1, MRPL4, S1PR2, DNMT1, EIF3G, P2RY11, ANGPTL6, SHFL, RDH8, OLFM2, PIN1, UBL5, FBXL12*
8	41.62 … 41.63	AMST, FR	*TMEM241, RIOK3, RBBP8*
11	24.14 … 25.76	SH	*SP2, PNPO, PRR15L, CDK5RAP3, COPZ2, NFE2L1, CBX1, SNX11, SKAP1, HOXB1, HOXB2, HOXB3, HOXB4, HOXB5, HOXB8, HOXB9, HOXB13, TTLL6, CALCOCO2, ATP5MC1, UBE2Z, SNF8, IGF2BP1, B4GALNT2, GNGT2, ABI3, PHOSPHO1, ZNF652, PHB1, NXPH3, SPOP, FAM117A, KAT7, TAC4*
17	21.06 … 22.28	TR	*TRIM13, SPRYD7, KPNA3, EBPL, ARL11, RCBTB1, PHF11, SETDB2, CAB39L, CDADC1, MLNR, FNDC3A, CYSLTR2*
18	68.39 … 69.18	AMST	*TMEFF2*
22	17.40 … 18.31	FR	*FERMT1, LRRN4, CRLS1, MCM8, TRMT6, CHGB, SHLD1, GPCPD1, PROKR2*

## Data Availability

Data will be made accessible from corresponding authors upon reasonable request.

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
