# Peer review of "The Genetic Diversity of Stallions of Different Breeds in Russia"

_genes, 2023, doi:10.3390/genes14071511_

Round 1

Reviewer 1 Report

Thank you for the great work on horse breeding. Here are small points to correct and improve.

Introduction:There is no continuity in the ideas between paragraph 1 and 2 (line 51-52)

Discussion: line177 incomplete sentence.

You mention the history of origin of breed in line 187 and line 188 would you give more details about this origin ?

line 208 talked about arabian horse unique feature what do you mean or what is the unique feature?

Author Response

Dear Reviewer,

Thank you for your consideration of this manuscript.

Introduction:There is no continuity in the ideas between paragraph 1 and 2 (line 51-52)

We added «For selection work, it is important to know the genetic characteristics of the breeds».

Discussion: line177 incomplete sentence.

«Using specially designed chips, inbreeding levels can be compared between breeds [6] as well as within individual groups of» is changed to  «Using custom designed chips, inbreeding levels can be compared both between breeds [6] and within individual groups [23].»

You mention the history of origin of breed in line 187 and line 188 would you give more details about this origin ?

We added « Other researchers have an idea of the genetic characteristics and population structure of the Asil, Caspian, Dareshuri, Kurdish and Turkmen horses as local Iranian horse breeds. (Salek Ardestani S, Zandi MB, Vahedi SM, Janssens S. Population structure and genomic footprints of selection in five major Iranian horse breeds. Anim Genet. 2022 Oct;53(5):627-639. doi: 10.1111/age.13243. For example, these are breeds, in the breeding of which the use of producers of other breeds is not allowed, the so-called purebred and half-breed breeds.

line 208 talked about arabian horse unique feature what do you mean or what is the unique feature?

We added «The genetic diversity of Arabian horses in Russia is based on the use of different breed types in breeding (koheilan, koheilan-siglawi, siglawi, hadban)»

Reviewer 2 Report

The authors presented data obtained from genetic analysis of 21 thoroughbred and semibreed horse breeds to search for heterozygous and homozygous regions.

Line 98: should be “509,617”

Figure 2: the font in the figure is too small, which is difficult to read

Line 135: “A high level of inbreeding is observed in standardbred horses.” where can the readers obtain this result?

Line 140: please add the standard deviation

The findings are limited. this paper can resubmit as short communication, or add some analyses based on the datasets. Why the authors did not detect and annotate the ROH hotspot? 

Minor editing of English language required, please have a check. 

Author Response

Dear Reviewier,

Thank you for your consideration of this manuscript.

Responses to comments are provided below.

Line 98: should be “509,617”

We corrected it

Figure 2: the font in the figure is too small, which is difficult to read

We corrected it

Line 135: “A high level of inbreeding is observed in standardbred horses.” where can the readers obtain this result?

We added information on localization of homozygous regions in standardbred trotters

Line 140: please add the standard deviation

We added information to the table

The findings are limited. this paper can resubmit as short communication, or add some analyses based on the datasets. Why the authors did not detect and annotate the ROH hotspot?

We added ROH localization analysis and annotated genes to results and discussion.

Round 2

Reviewer 2 Report

Accept in present form